# E-Cigarette Quit Attempts and Experiences in a Convenience Sample of Adult Users

**DOI:** 10.3390/ijerph20032332

**Published:** 2023-01-28

**Authors:** Meagan A. Bluestein, Geronimo Bejarano, Alayna P. Tackett, Jaimie C. Duano, Shelby Grace Rawls, Elizabeth A. Vandewater, Jasjit S. Ahluwalia, Emily T. Hébert

**Affiliations:** 1Michael & Susan Dell Center for Healthy Living, Austin Campus, School of Public Health, University of Texas Health Science Center at Houston (UTHealth), Austin, TX 78701, USA; 2Center for Tobacco Research, The Ohio State University, Columbus, OH 43210, USA; 3Department of Internal Medicine, Division of Medical Oncology, The James Comprehensive Cancer Center, The Ohio State University, Columbus, OH 43210, USA; 4Population Research Center, University of Texas at Austin, Austin, TX 78701, USA; 5Department of Health Promotion and Behavioral Sciences, Austin Campus, School of Public Health, The University of Texas Health Science Center at Houston (UTHealth), Austin, TX 78701, USA; 6Department of Behavioral and Social Sciences, Brown University School of Public Health, Providence, RI 02912, USA; 7Legorreta Cancer Center, Brown University, Providence, RI 02912, USA

**Keywords:** vaping, cessation, interventions

## Abstract

Most e-cigarette users report planning to quit, but there is a paucity of evidence-based interventions for e-cigarette cessation. In the absence of interventions for e-cigarette cessation, we sought to understand how and why e-cigarette users attempt to quit on their own. Participants were recruited from Amazon Mechanical Turk, an online crowdsourcing platform. Those who reported they had ever used e-cigarettes regularly and had attempted to quit e-cigarette use were eligible for participation. Measures included demographic characteristics, other tobacco product use, e-cigarette device characteristics, barriers to quitting e-cigarettes, and facilitators to quitting e-cigarettes. A content analysis was conducted on twotwo open-ended questions that asked about advice respondents had for others trying to quit vaping and resources they wished they had during their quit attempt. Descriptive analyses were performed (means/standard errors; frequencies/proportions). A total of 89.0% reported using an e-cigarette with nicotine, 20.2% reported a nicotine concentration of 4–6 mg/mL%, 32.8% reported using multiple flavors, and 77.7% reported using their e-cigarette every day or some days. The primary reason reported for wanting to quit e-cigarettes was health concerns (42.2%), and 56.7% reported trying to quit “cold turkey”. During quit attempts, 41.0% reported intense cravings and 53.1% reported stress as a trigger. From the content analysis, the most commonly cited suggestion for those wanting to quit e-cigarettes was distractions/hobbies (19.9%), followed by reducing/tapering down nicotine (16.9%). Descriptive information on demographics, e-cigarette use, device characteristics, barriers, facilitators, and quit methods provides a first step in identifying factors that contribute to successful interventions designed for e-cigarette cessation.

## 1. Introduction

In the United States, an estimated 5.1% of adults aged 18 and older were current (past 30 days) e-cigarette users in 2020 [1]. A study using repeated cross-sectional data from the nationally representative Behavioral Risk Factor Surveillance System found that the prevalence of current e-cigarette use increased from 2017 to 2018 and decreased slightly in 2020; however, the prevalence of daily e-cigarette use has consistently increased over the last few years [1]. While it is generally recognized that e-cigarettes are less harmful compared with conventional cigarettes, the majority of e-cigarettes sold in the U.S. contain nicotine [2], and may expose users to nicotine levels proportionate to or higher than combustible cigarettes [3]. The latest generation of nicotine salt-based e-cigarette devices deliver nicotine at substantially higher levels than earlier e-cigarette devices [4,5], making nicotine dependence a serious concern, particularly in otherwise tobacco naïve users. In a 2016–2018 analysis of the Population Assessment of Tobacco and Health (PATH) study, researchers found that current established e-cigarette users (had ever used fairly regularly, now uses every day or some days) had a nicotine dependence level of 1.95 (out of 5), and 13.3% had attempted to quit using e-cigarettes in the past year [6]. 

Nearly two-thirds of adult e-cigarette users in the wave 3 2015–2016 PATH study reported plans to quit e-cigarettes for good [7], yet to date, there are few evidence-based interventions specifically designed for individuals who want to quit vaping. E-cigarette users offer multiple reasons for wanting to quit, such as health and safety concerns, financial cost, and a desire to be free from addiction [8,9]. Few studies have explored how e-cigarette users attempt to quit vaping, and what their quit experiences are like. A cross-sectional survey of dual e-cigarette and combustible cigarette users found that e-cigarette users had a lifetime median of five e-cigarette quit attempts, and that the most common methods used were cutting down (68%), advice from a doctor (28%), and quitting “cold turkey (24%)” [10]. Preliminary evidence suggests that the e-cigarette cessation experience may share some similarities with quitting conventional cigarettes. In a recent analysis of the Wave 2 PATH survey, 40% of U.S. adult exclusive e-cigarette users reported withdrawal symptoms when they tried to stop or reduce e-cigarette use, and unsuccessful quitters appeared to have a greater number of withdrawal symptoms than successful quitters [11].

Given the diversity of e-cigarette device types, product characteristics (e.g., nicotine type, nicotine strength), and the number of e-cigarette users who want to quit, research is needed to inform effective vaping cessation interventions. In the absence of evidence-based treatments specific to e-cigarette cessation [12], we sought to understand how and why e-cigarette users are attempting to quit on their own. The purpose of the present study is to describe the use patterns and quit experiences of past or current regular e-cigarette users who made at least once quit attempt.

## 2. Materials and Methods

Participants were recruited from Amazon Mechanical Turk (MTurk), an online crowdsourcing platform with data collected in June 2021. All study procedures were approved by the University of Oklahoma Health Sciences Center Institutional Review Board (IRB). 

### 2.1. Inclusion Criteria & Final Sample

A total of 1137 participants met the study inclusion criteria: (1) English speaker, (2) age 18 or older, (3) ever used e-cigarettes regularly (i.e., daily, weekly, or monthly at some point in their life), and (4) made at least one lifetime e-cigarette quit attempt. Because of concerns regarding the quality of MTurk data, we closely examined the pattern of participant responses, and dropped those who exhibited implausible, dubious, or otherwise untrustworthy response patterns. This is an extremely conservative approach to data quality and control, but given the known issues with MTurk data, we believe it was warranted to foster confidence in the quality of the data retained in the analytic sample. On the basis of data review, n = 350 were dropped from analyses for a variety of non-completion reasons (n = 190 answered screener questions only, n = 89 finished in less than 5 min (the lower 10% of the distribution of time spent on the survey, median time to finish the survey was 9.6 min), n = 36 didn’t complete the survey at all, n = 34 reported different ages on the screener vs. the survey, and n = 1 did not agree to provide accurate data). The final sample included n = 787 adult participants ages 18 and over who, at some point(s) in their lifetime, were regular e-cigarette users and had made at least one attempt to quit e-cigarette use. 

### 2.2. Measures 

#### 2.2.1. Demographics 

Participants responded to a variety of demographic questions including age (in years), gender (men, women, transgender, non-binary, genderfluid/genderqueer, other), race (White, Black/African American, Asian, other or multi-race), ethnicity (Hispanic or non-Hispanic), annual household income (less than $25,000, $25,000-$49,999, $50,000-$99,999, $100,000 or more), and educational attainment (less than high school, high school graduate/GED, some college/Associate’s degree, Bachelor’s degree, and Master’s/professional degree/PhD). 

#### 2.2.2. E-Cigarette Use and Device Characteristics 

Participants indicated how often they currently use e-cigarettes (every day, some days, not at all), the timing of their last e-cigarette use (earlier today, within the past 7 days, within the past 30 days, more than 30 days ago), how frequently they use e-cigarettes in a single day (0–4 times per day, 5–9 times per day, 10–14 times per day, 15+ times per day, where a single “time” was defined as about 15 puffs or lasting about 10 min), and 2 items from the validated Penn State Electronic Cigarette Dependence Scale [13]: how soon after waking they first use their e-cigarette (0–5 min, 6–15 min, 16–30 min, 31–60 min, more than 60 min), the number of nights per week they wake from sleep to use their e-cigarette (0–1 nights, 2–3 nights, 4+ nights), and if have they completely quit using e-cigarettes (yes/no). Participants also indicated the type of device they use most often (disposable, prefilled pods/cartridges, tank refilled with liquids, customizable mod system), and their usual nicotine concentration (1–3 mg/mL, 4–6 mg/mL, 7–12 mg/mL, 13–17 mg/mL, 18–24 mg/mL, 25–39 mg/mL, 40–49 mg/mL, 50+ mg/mL, don’t know).

#### 2.2.3. Other Tobacco Product Use 

In addition to the timing of last cigarette use (earlier today, within the past 7 days, within the past 30 days, more than 30 days ago, never), lifetime ever use (yes/no) of 9 tobacco products (cigar products, hookah, smokeless tobacco, roll-your-own cigarettes, pipe, snus, dissolvable tobacco, bidis, and heat-not-burn products) was measured. 

#### 2.2.4. E-Cigarette Quitting: Reasons for Quitting, Methods Tried, and Experiences during Quit Attempts

Participants indicated whether their reasons to stop using e-cigarettes included any of the following (yes/no for each): cost, dislike of taste or side effects, e-cigarette use did not aid conventional cigarette quit attempts, poor quality/defective/break easily/a hassle to use, health risk concerns, just experimenting, or some other reason.

They also indicated how many different quit methods they had tried (cold turkey, decreasing nicotine content, trying to use less often, switching flavors, nicotine replacement therapy, other), as well as resources they used to aid quitting (a phone app, a free text messaging program, a telephone quit line, resources from a website, other).

Participants were asked who they would trust to give them advice about quitting e-cigarettes (doctors or other medical professionals, scientists or researchers, family members or friends, vape store employees, others who have quit vaping, and other). 

During quit attempts, participants indicated whether or not they had experienced any of 12 possible withdrawal symptoms (intense cravings, tingling in hands and feet, sweating, nausea, headaches, coughing, insomnia, difficulty concentrating, anxiety, depression, weight gain, and other), any of 5 possible triggers (cravings, seeing someone else vape, drinking alcohol, stress, anxiety, other) when attempting to quit e-cigarettes, and whether they had used any of 5 possible coping methods in response to triggers (distraction, meditation or breathing exercises, eating or chewing gum or ice, talking to a family member or friend, other). 

#### 2.2.5. Content Coding of Open-Ended Responses Regarding Quitting E-Cigarettes 

Two open-ended, optional questions were included in the survey regarding advice respondents had for others trying to quit (“What advice would you have for someone trying to quit vaping?”), as well as resources that they wished they had during their own quit attempt (“What resources do you wish you had to help you quit vaping”).

Four independent coders reviewed the open-ended responses and developed a codebook to reflect participant responses to the two questions. Responses were coded for mentions of specific quit strategies used (i.e., cold turkey, medications, distractions/hobbies, reducing or tapering down on nicotine concentration, avoidance), mentions of quitting resources used (i.e., professional help, web/mobile resources, peer resources), as well as mentions of social support for quitting, commitment/motivation to quit, and negative health effects of vaping. Coders trained with three rounds of practice coding (60 responses total) until interrater reliability was reached for each category (Cronbach’s alpha of at least 0.8), before coding the entire sample. Responses that were irrelevant, nonsensical, or were suspected of being copied from an online resource (e.g., identical language and syntax to Smokefree.gov) were excluded from analysis.

### 2.3. Analysis

Because so little is known about e-cigarette quit attempts, our primary purpose in this study is to provide some initial descriptive statistics regarding the e-cigarette, tobacco use, and quit attempt experiences among e-cigarette users who have tried to quit using e-cigarettes. Means and standard deviations or frequencies and proportions, respectively, were used to describe continuous or categorical variables as appropriate. Analyses were conducted in SAS version 9.4-TSlevel1M6 [14].

## 3. Results

Recall that based upon the inclusion criteria, sample participants were English speaking adults ages 18 and older who reported using e-cigarettes regularly at some point in their life and had tried to quit using e-cigarettes at least once. 

### 3.1. Demographics 

Sample demographic characteristics are shown in Table 1. Participants were 37.4 years old on average, 21% of the sample were young adults (ages 19–29), and the sample age ranged widely, from 19–70 years old. The sample was largely male (59%), White (76.4%), non-Hispanic, (86.5%), and college educated, with 47.4% holding a bachelor’s degree. This may reflect the characteristics of MTurk participants, of individuals meeting the study inclusion criteria, or both.

### 3.2. E-Cigarette Use and Device Characteristics 

Table 2 shows the summary statistics for e-cigarette use and device characteristics. A total of 72.3% of respondents reported using e-cigarettes within the last 30 days, and 42.6% (n = 335) reported that they had completely quit using e-cigarettes. However, there were discrepancies among those who reported they had completely quit using e-cigarettes, as 18.3% (n = 144) of self-reported “quitters” also reported using e-cigarettes in the past 30 days, and 21.2% reported that they now use e-cigarettes every day or some days (n = 167). This may indicate the need to carefully define the meaning of quitting in future studies, as participants may think of “completely quitting” as an episodic, rather than permanent state or consider themselves to be “quitters” even after a short period of time. 

A majority of the sample used e-cigarettes some days or every day (77.7%), and roughly two thirds (67.2%) reported using e-cigarettes within 30 minutes of waking. The vast majority of participants (89.0%, n = 700) reported using e-cigarettes with nicotine, but almost one-quarter (23.5%, n = 185) reported that they did not know the nicotine concentration. The mode (20.2%) for nicotine concentration was 4–6 mg/mL%. As this concentration is rather low, it is possible that this sample of users consumes a lower than average amount of nicotine when vaping, or that they tend to underestimate the nicotine concentration they typically consume while vaping e-cigarettes. Regardless, this issue warrants careful examination in future work, as nicotine consumption has a large influence on the ability of users to quit [15,16].

More than half of participants were dual e-cigarette/cigarette users, with 53.4% reporting they had used e-cigarettes and cigarettes in the past 30 days, while 18.9% had used e-cigarettes exclusively in the past 30 days. It is worth noting that respondents were asked if they had ever tried any of the other tobacco products (e.g., dissolvables, cigars, etc.) during their lifetime, even once. Thus, an individual whose cigar use was limited to a single puff of a friend’s cigar could answer in the affirmative, which might explain the relatively high percentage of ever users for some products (e.g., 76.4% for any cigar product).

### 3.3. E-Cigarette Quitting: Reasons for Quitting, Methods Tried, and Experiences during Quit Attempts

Table 3 reports summary statistics for reasons to quit, strategies used to quit using e-cigarettes, and experiences during quitting. Of the adult current e-cigarette users and e-cigarette quitters, 83.7% had attempted to quit at least once in the past year, 42.2% reported the main reason they stopped/wanted to stop using e-cigarettes was due to health concerns, 62.8% said they would trust doctors or other medical professionals to give them advice on quitting e-cigarettes, and 56.7% reported quitting cold turkey as a method used to attempt quitting e-cigarettes. 

During attempts to quit or cut back, 41.0% reported intense cravings as a side effect, 53.1% reported stress as a trigger that made them want to use again, 67.0% reported distraction as a method to cope with triggers, 22.4% reported using website resources to help them quit, while 54.9% reported not using any resources at all.

### 3.4. Content Coding of Open-Ended Responses Regarding Quitting E-Cigarettes

A total of 638 (81.1% of the sample) open-ended responses were coded for analysis. The most commonly cited suggestion for those wanting to quit e-cigarettes was distractions, hobbies, or alternative activities (e.g., exercise, meditation, 19.9%), followed by reducing or tapering down on nicotine (16.9%), medications or nicotine replacement therapy (14.1%), and peer resources (i.e., other vapers who have quit or support groups, 13.8%). Other resources mentioned included professional help (i.e., help from a physician or mental health professional, 12.7%), and web or mobile resource (i.e., app, website, or online forum, 11.3%). Aside from specific quit strategies and resources mentioned, mentions of commitment, motivation, and/or willpower were common (21%), as were mentions of social support from friends or family (19.7%), and the negative health effects of vaping (13.0%).

## 4. Discussion

The present study summarizes the experiences of a convenience sample of adult e-cigarette users who have attempted to quit e-cigarettes. In the current sample, a majority of respondents (77.6%) reported current e-cigarette use on some days or every day, most reported using pod-based devices, and 83.7% had attempted to quit vaping at least once in the past year. Notably, most of the sample (60.9%) reported conventional cigarette use in the past 30 days. The main reasons that participants wanted to quit using e-cigarettes were “I was concerned about the health risks caused by them” (42.2%) and “They cost too much money” (22.6%). These results are consistent with a prior qualitative study among a sample of young adult e-cigarette users [8], in which general health (29.8%) and financial cost (26.5%) were the most common reasons cited for wanting to quit. 

Quitting “cold turkey” was the most popular quit method endorsed by participants. This is in contrast to a content analysis of a quit vaping community on Reddit, which found that 66.9% of community members preferred a gradual reduction or by tapering the nicotine content, compared to 33.1% of those who preferred the “cold turkey” approach [15]. It may be possible that users’ preferences or quit method depends on the type of e-cigarette device that they use. While tapering down on nicotine content gradually may be possible with tank or mod-type devices, it may be less feasible with closed system devices like disposables, pods, or cartridges. In addition, quit methods may vary based on tobacco use status. Importantly, most e-cigarette users in this sample were also current users of conventional cigarettes (dual users). Given that 18.8% of e-cigarette users in this study claimed that they wanted to quit using e-cigarettes because they did not help with cigarette cessation, it is possible that this is how they became dual users. This finding agrees with a previous study of dual users of e-cigarettes and cigarettes from the wave 4 (2016–2018) and wave 5 (2018–2019) PATH study, which found that 16.0% reported using e-cigarettes for quitting smoking and harm reduction; however, this was not associated with cigarette cessation one year later [16]. In a similar cross-sectional survey of adult dual cigarette and e-cigarette users, the methods respondents used to quit conventional cigarettes were highly correlated with methods they used to quit e-cigarettes, such as nicotine replacement therapy and cutting back [7]. In the same study, over 20% of the sample increased use of conventional cigarettes during their attempt to quit vaping [12]. Thus, intervention strategies for e-cigarette cessation need to carefully consider users’ baseline tobacco use status, and consider approaches that minimize harm from conventional cigarette smoking in the effort to achieve abstinence from all nicotine and tobacco products.

The results of this study have several implications for future research and development of interventions for e-cigarette cessation. First, a majority of users indicated that they would trust advice for vaping cessation from a medical professional, followed by advice from family members of friends. Future interventions or communication strategies might therefore consider medical settings (e.g., brief interventions during a primary care visit) for intervention delivery, and/or incorporating support from family or friends. Second, many respondents reported experiencing side effects or withdrawal symptoms during e-cigarette quit attempts, such as headaches and intense cravings. Future intervention strategies might consider incorporating gold standard smoking cessation treatments including pharmacotherapy and nicotine replacement therapy to help alleviate these symptoms. Finally, the discrepancies in answers to e-cigarette use status from different questions about e-cigarette use and quitting (i.e., 18.3% of those who consider themselves quitters have used in the past 30 days) indicate a need to clearly define quitting with specific criteria, including time since last use. Future research should examine users’ perceptions of what it means to be “completely quit”, and if self-identifying as a quitter despite recent use is associated with intentions or motivation to quit. 

## 5. Strengths and Limitations

One limitation of note is that using a convenience sample from Amazon mTurk may severely limit the generalizability of findings to the wider U.S. population [17]. It should be noted, however, that while mTurk participants differ from the U.S. population demographically, smoking prevalence among mTurk users is comparable to data from the National Adult Tobacco Survey [18]. Moreover, given the known issues with the quality generated by MTurk respondents, we undertook a variety of steps to screen and process the data, fostering confidence in the quality of data used in our analyses. Indeed, the value of these data and analyses are elevated in the context of almost no investigation into e-cigarette users quit attempts and experiences. Another limitation is that this sampling approach offers only respondent self-report, as it is not possible to objectively confirm use or quit status. Finally, these data were collected during the COVID-19 pandemic, which may have impacted e-cigarette use and quitting behaviors.

## 6. Conclusions

This study is the among the first to conduct an examination focused on e-cigarette users who had all made a quit attempt. To date, much of what we know is based on samples of tobacco users, who use e-cigarettes as a way to quit using combustibles, which is an important avenue for harm reduction. However, data from younger cohorts indicates that an increasing number of young adults use e-cigarettes only. A sizeable portion of these young adults have become addicted to e-cigarettes, and are indicating their desire to quit. Descriptive information regarding demographic characteristics, e-cigarette use and device characteristics, reasons, methods, and resources used to quit e-cigarettes specifically, as well as the experience of users during a quit attempt (cravings, withdrawal symptoms, triggers, etc.) is a critical first step to developing effective interventions for those who wish to quit using e-cigarettes.

## Figures and Tables

**Table 1 ijerph-20-02332-t001:** Sample Demographic Characteristics.

		n	%
Age	Mean (SD)	37.4 (10.3)	
Age group	18–24	48	6.1
	25–34	336	42.7
	35–44	229	29.1
	45–54	110	14.0
	55–64	49	6.2
	65+	15	1.9
Gender	Women	317	40.3
	Men	464	59.0
	Transgender/Other	6	0.8
Race	Asian	36	4.6
	Black or African American	105	13.3
	White	601	76.4
	Other, Multi-race	40	5.1
	Did not respond	5	0.6
Ethnicity	Hispanic	106	13.5
	Non-Hispanic	681	86.5
Annual Household Income	Less than $25,000	104	13.2
	$25,000–$49,999	248	31.5
	$50,000–$99,999	324	41.2
	$100,000 or more	104	13.2
	Did not respond	7	0.9
Educational Attainment	High school graduate, GED	74	9.4
	Some college/Associate’s degree	203	25.8
	Bachelor’s degree	373	47.4
	Master’s, Professional degree, PhD	137	17.4

Note: Total N = 787.

**Table 2 ijerph-20-02332-t002:** E-cigarette use and device characteristics.

		n	%
Last e-cigarette use	Earlier today	310	39.4
	Within the past 7 days	188	23.9
	Within the past 30 days	71	9.0
	More than 30 days ago	218	27.7
Current e-cigarette use	Every day or some days	611	77.6
	Not at all	176	22.4
Has completely quit using e-cigarettes	Yes	335	42.6
	No	452	57.4
Frequency of daily e-cigarette use	0–4 times per day	305	38.8
	5–9 times per day	233	29.6
	10–14 times per day	124	15.8
	15+ times per day	125	15.9
E-cigarette nicotine concentration	No nicotine	87	11.1
	1–3 mg/mL	107	13.6
	4–6 mg/mL	159	20.2
	7–12 mg/mL	79	10.0
	13–17 mg/mL	31	3.9
	18–24 mg/mL	48	6.1
	25–49 mg/mL	45	5.8
	50+ mg/mL	46	5.8
	I don’t know the concentration	185	23.5
E-cigarette device type	Disposable	205	26.1
	Pre-filled pods or cartridges	400	50.8
	Refillable tank	121	15.4
	Customizable mod system	60	7.6
Time until first e-cigarette upon waking	0–5 min	165	21.0
	6–15 min	199	25.3
	16–30 min	165	21.0
	31–60 min	110	14.0
	60 min+	148	18.8
Nights per week wake from sleep to use e-cigarette	0–1 nights	586	74.5
	2–3 nights	171	21.7
	4+ nights	30	3.8
Last combustible cigarette use	Earlier today	258	32.8
	Past 7 days	152	19.3
	Past 30 days	69	8.8
	More than 30 days ago	300	38.1
	Never	8	1.0
Dual e-cigarette/cigarette use in the past 30 days	Dual use	420	53.4
	E-cigarettes only	149	18.9
	Cigarettes only	59	7.5
	Neither	159	20.2
Ever use of other tobacco products *	Any cigar product	601	76.4
	Hookah	372	42.3
	Smokeless tobacco	251	31.9
	Roll-your-own cigarettes	298	37.9
	Pipe	174	22.1
	Snus	141	17.9
	Dissolvable tobacco	33	4.2
	Bidis	63	8
	Heat-not-burn products	78	9.9

Note: Total N = 787. * Responses were “check all that apply”, and thus do not equal 100%.

**Table 3 ijerph-20-02332-t003:** E-cigarette Quit Attempts Reasons, Methods, and Experience.

		%	
Number of E-Cigarette Quit Attempts in Past Year	Did not attempt to quit in past year	128	16.3
	1 time	123	15.6
	2 times	197	25.0
	3–5 times	237	30.1
	6–9 times	44	5.6
	10 or more times	58	7.4
Main reason for quitting	Cost	178	22.6
	Unpleasant taste or side-effects	48	6.1
	Did not aid cigarette cessation	148	18.8
	Poor quality, hassle to use	42	5.3
	Health risks	332	42.2
	Just experimenting	18	2.3
	Other	21	2.7
Person of trust for quit advice *	Doctors, other medical professionals	494	62.8
	Scientists, researchers	329	41.8
	Family members, friends	353	44.9
	Vape store employees	41	5.2
	Others who have quit vaping	232	29.5
	Other	11	1.4
Quit methods used *	Cold Turkey	446	56.7
	Decrease nicotine content	299	38.0
	Use less often	440	55.9
	Change flavor	92	11.7
	Nicotine replacement therapy	120	15.3
	Other	13	1.7
Side effects or withdrawal symptoms experienced *	Intense cravings	323	41.0
	Tingling in hands and feet	58	7.4
	Sweating	131	16.7
	Nausea	76	9.7
	Headaches	237	30.1
	Coughing	59	7.5
	Insomnia	77	9.8
	Difficulty concentrating	139	17.7
	Anxiety	184	23.4
	Depression	72	9.2
	Weight gain	53	6.7
Triggers experienced *	Cravings	411	52.2
	Seeing someone else vape	298	37.9
	Drinking alcohol	206	26.2
	Stress	418	53.1
	Anxiety	277	35.2
	Other	23	2.9
Trigger coping methods used *	Distraction	527	67.0
	Meditation or breathing exercises	246	31.3
	Eating or chewing gum or ice	382	48.5
	Talking to a family member or friend	148	18.8
	Other	33	4.2
Resources used to quit *	An app for my phone	135	17.2
	A free text messaging program	104	13.2
	A telephone or quit line	87	11.1
	Resources from a website	176	22.4
	None of the above	432	54.9

Note: Total N = 787. * Responses were “check all that apply”, and thus do not equal 100%.

## Data Availability

The data presented in this study are available on request from the corresponding author.

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
