# Peer review of "E-Cigarette Quit Attempts and Experiences in a Convenience Sample of Adult Users"

_ijerph, 2023, doi:10.3390/ijerph20032332_

Round 1
Reviewer 1 Report
The authors have touched upon a very important aspect of quitting e cigarette. The manuscript is well written and conveys clear conclusion. There are a few points that I want to draw the attention of the authors
1. As described by authors this is a self reported quit attempt by the respondents and could not be verified
2. Some questions on users attitude - whether they consider e cigarette as/more/less harmful than conventional cigarette would be useful.
3. The sampling method could have been improved. The authors should attach a supplementary file of all the initial responses recieved and describe in more detail how they excluded some from it only. The methodology of exclusion can be explained in more detail.
Reviewer 2 Report
The authors surveyed those adults who ever used e-cigarettes regularly and had attempted to
quit e-cigarette use by using Amazon Mechanical Turk. Based on the final 787
samples, the manuscript shows the frequencies and proportions of the demographic
characteristics, e-cigarette/cigarette use, and device characteristics. It also
analyzed the e-cigarette quit attempts reasons, methods, and experiences among
all ever e-cig users. The
survey design is meaningful for studying e-cig use and cessation. However, the literature
review and background introduction can
be improved by adding more US e-cig studies, and the results may have some
problems. My 17 comments are attached. Please check.
